# Impact of Post-Harvest Apple Scab on Peel Microbiota, Fermentation Dynamics, and the Volatile/Non-Volatile Composition of Cider

**DOI:** 10.3390/molecules30112322

**Published:** 2025-05-26

**Authors:** Valeria Gualandri, Roberto Larcher, Elena Franciosi, Mauro Paolini, Tiziana Nardin, Ilaria Pertot, Raffaele Guzzon

**Affiliations:** 1Centro di Trasferimento Tecnologico, Fondazione Edmund Mach, Via Mach 1, 38010 San Michele all’Adige, TN, Italy; valeria.gualandri@fmach.it (V.G.); mauro.paolini@fmach.it (M.P.); tiziana.nardin@fmach.it (T.N.); raffaele.guzzon@fmach.it (R.G.); 2Center Agriculture Food Environment (C3A), University of Trento, Via Mach 1, 38010 San Michele all’Adige, TN, Italy; ilaria.pertot@unitn.it; 3Centro di Ricerca e Innovazione, Fondazione Edmund Mach, Via Mach 1, 38010 San Michele all’Adige, TN, Italy; elena.franciosi@fmach.it

**Keywords:** apple scab, post-harvest, cider, apple microbiota, circular economy, food remediation

## Abstract

Apple scab is a disease caused by *Venturia inaequalis*; it alters the vegetative cycle of apple trees and affects the fruits in orchards or during post-harvest storage. Utilizing rotten apples in cidermaking is a promising technique to mitigate crop losses; nonetheless, uncertainties persist regarding the beneficial effects of damaged fruits. This study involves a thorough chemical analysis of cider produced from both healthy and scab-infected fruits to identify compositional changes caused by microbial proliferation and to assess their impact on cider quality. Apples infected by post-harvest apple scab, as opposed to uninfected apples, were employed in cidermaking. The peel microbiota was described by plate count, and next-generation sequencing-based metabarcoding methods were used to describe the peel microbiota, while HPLC and GC MS-MS were used to characterize the cider compositions. Apples infected with post-harvest scab host a specific fungal consortium with higher biodiversity, as evidenced by the Shannon evenness index, especially in the fungi kingdom. The presence of apple scab slows fermentation by up to 23%, lowers ethanol accumulation by up to 0.4%, and affects certain cider constituents: sugars, alcohols, amino acids, fatty acids, and esters. The statistical treatment of data relative to the chemical profile (PLS and PCA on the 31 compounds with VIP > 1) distinguishes ciders made from altered or safe fruits. Scab-infected apples can be valorized in the agri-food industry; however, microbiota alterations must not be underestimated. It is necessary to implement adequate mitigation strategies.

## 1. Introduction

Apple scab, caused by *Venturia inaequalis* (Cooke) Wint., is one of the most harmful diseases of apple trees, present in all geographical areas where apples are grown, with a prevalence in temperate countries characterized by cool and wet climates in early spring [1]. The disease can significantly reduce production, with economic losses reaching up to 70% of crop value. It is characterized by a wide range of symptoms affecting nearly all the organs of the apple tree during the entire growing season [2]. Olive-green to dark brown lesions appear on leaves, and they can result in curling, deformation, and premature drop, thus reducing photosynthesis. *V. inaequalis* causes dark corky lesions on fruits, which can crack and deform them, thus lowering their marketability. The disease can also affect flowers, stems, petioles, and buds, resulting in flower abortion, fruit drop, and a consequent reduction in yield. All secondary infections commence with conidia, which are single-celled, brown, elliptical, or oval-shaped structures, typically measuring 6–12 μm in width and 12–22 μm in length. Conidia are produced at the tips of short hyphae on the velvety surface of the lesions [3]. Under favorable conditions, conidia germinate, producing germ tubes and hyphae that infiltrate host tissues to initiate new infections. Late fruit infections may not be visible at harvest; nevertheless, during storage, they can manifest as black spots [4] and result in additional losses [5]. Post-harvest infections may also facilitate the proliferation of other pathogenic fungi [4]. Ensuring consistent fruit quality is consequently more challenging due to the additional labor and equipment costs associated with sorting and removing symptomatic fruits, which are typically discarded and composted.

The disease can be managed using an integrated approach that combines the selection of a suitable growing site, resistant varieties [6], and the application of fungicides [2], biocontrol agents, and natural substances [7]. However, these measures are not always sufficient to guarantee the absence of symptoms on fruits, both at harvest and after post-harvest storage. Alternative strategies to valorize unmarketable fruits are therefore essential for agricultural sustainability and the reduction of food waste. A promising method is the production of fermented beverages, such as cider. Fermentation provides several advantages, including the prolongation of apple juice shelf life due to the accumulation of ethanol and/or organic acids, as well as the reduction of carbon sources that could facilitate the proliferation of spoilage or pathogenic microbes [8]. Additionally, fermentation enhances sensory qualities [9], may produce nutraceutical compounds [10,11], and detoxifies environmental pollutants or the proliferation of pathogenic microorganisms. For example, *Saccharomyces cerevisiae* can adsorb ochratoxin A during alcoholic fermentation [12] and has also shown potential as a bioremediation agent for heavy metal pollution [13]. Similarly, *Rhodotorula mucilaginosa* and *Diutina rugosa* may remove aflatoxin B1 and zinc, respectively [14]. *Lactobacillus* and *Enterococcus* spp. can eliminate up to 99% of heavy metals in water or food matrices [15,16].

Today, most fermented beverages are produced using selected cultures of yeast and/or bacteria to ensure a consistent fermentation process [17,18], often following pasteurization. However, due to the sensitivity of certain fruit components to heat, pasteurization is rarely utilized in high-value products, such as wine and cider [19]. In this case, the role of the wild microbiota in the fermentation process becomes significant. Naturally occurring yeast and bacteria that colonize fruits can shape the organoleptic profile of fermented beverages [20,21], increasing their recognizability and complexity, traits that are highly valued by consumers. Conversely, these microorganisms may also induce undesirable alterations [22]. Microbial colonization during the final stage of ripening or post-harvest reduces the fermentative potential of the raw materials [23], as microorganisms deplete nutrients or produce inhibitory compounds. These factors are extensively researched in the wine and beer industries [24,25], although there is insufficient information regarding their impact on cider production. Cider is traditionally produced in colder regions, such as northern Spain, France, and England, where climatic conditions favor the cultivation of apples [26], presenting a potential opportunity to utilize apples that are unmarketable due to post-harvest infections of *V. inaequalis*. However, the impact of apple scab on the fermentative potential and compositional profile of cider requires further investigation.

This study incorporates a thorough chemical characterization of cider produced from both healthy and scab-infected fruits to identify compositional changes caused by apple scab. Furthermore, we investigated the biodiversity of apple skin microbiota after post-harvest storage, examining both healthy and scab-infected fruits, using a combination of traditional microbial isolation and next-generation sequencing (NGS)-based metabarcoding techniques. These findings are particularly valuable for the industrial valorization of scab-infected fruits and support strategies that use unmarketable fruits for cider production while maintaining high product quality.

## 2. Results

### 2.1. Physicochemical Analysis of Apples, Progress of Alcoholic Fermentation, and Chemical Features of Ciders

In each cultivar, the physicochemical parameters of apples (weight, hardness, sugars, acidity, juiciness, and total polyphenols) exhibited no significant differences (Appendix A) between SF and RT fruits; the differences between GL and GA samples were consistent with the characteristics of the two different apple cultivars.

The fermentation of GL samples (Figure 1) presented a lag phase of 3 days. AF lasted 12 days with a final sugar residue of 0.59 g/L in SF and 0.67 g/L in RT samples (Table 1). The CO_2_ release and AF rate demonstrated statistically significant differences (one-way ANOVA, *p* < 0.05) between SF and RT samples starting on the 8th day of AF. A V_max_ of 2.35 ± 0.37 g L^−1^ Day^−1^ was reached in GO_SF (10th day of AF), while a V_max_ of 1.81 ± 0.10 g L^−1^ Day^−1^ was attained in GO_RT (12th day of AF). The presence of apple scab in GL apples affects the chemical features of the resulting cider (Table 1, Appendix A). Differences were found (one-way ANOVA, *p* < 0.05) in ethanol content (7.5% in GO_SF vs. 7.1% in GO_RT) and sugars, with a prevalence of fructose in SF samples and glucose in RT samples. In GA apples, AF occurred in 9 days (Figure 1), with a lag phase of 24 h. A V_max_ of 4.40 ± 0.17 g L^−1^ Day^−1^ was reached in SF samples compared to a V_max_ of 4.45 ± 0.44 g L^−1^ Day^−1^ recorded in RT samples. No differences were observed in the progression of AF until the 7th day, when a slowdown in sugar consumption in the RT samples resulted in lower ethanol accumulation (7.1% in GA_RT vs. 7.3% in GA_SF) and higher sugar residue (0.94 g/L in GA_RT vs. 0.75 g/L in GA_SF). GL tests indicated that cider produced from SF apples exhibited a prevalence of fructose, while cider made from RT fruits was characterized by a higher concentration of glucose. A total of 60 chemical compounds were identified and quantified in the ciders (Table 2). The PLS regression analysis was conducted using the complete dataset, revealing a direct correlation between cider composition and the presence of apple scab (Figure 2).

The PLS regression identified 31 compounds with a variable importance projection score (VIP) greater than 1: 14 amino acids (glycine, L-histidine, L-serine, L-valine, L-tyrosine, L-alanine, L-threonine, L-tryptophan, L-methionine, L-isoleucine, L-leucine, L-aspartic acid, L-glutamic acid, and L-phenylalanine), 4 fatty acids (butanoic, hexanoic, nonanoic, and valeric acid), 4 esters (ethyl butyrate, ethyl hexanoate, *n*-hexyl acetate, and 2-phenylethanol), 3 aromatic compounds (guaiacol, eugenol, and benzyl alcohol), 3 alcohols (ethanol and 3-methyl-2-butanol), 2 terpenes (linalool and geraniol), and 1 sugar (fructose). These 31 compounds were considered during the PCA.

Figure 3 represents the plane based on factors 1 and 2 (F1 and F2), which collectively account for 87.8% of the total variability. PCA effectively differentiated between ciders made from the two apple cultivars, with GL cider positioned in the quadrants with negative F2 values and GA cider in the quadrants with positive F2 values. Additionally, PCA separated those ciders produced from SF apples (F1 > 0) and those obtained from RT apples (F1 < 0). The F1 component, which accounts for 74.4% of the total variability, primarily differentiates between SF and RT ciders, while F2 represents 18.0% of the variability, distinguishing the cultivars.

Ciders made from SF apples had a higher concentration of volatile compounds of fermentative origin, including esters, and, in the case of GL samples, a notable concentration of terpenes. These ciders also had higher ethanol and fructose levels. Conversely, ciders produced from RT apples were characterized by a higher amino acid content.

### 2.2. Analysis of the Apple Peel Microbiota in Healthy and Scab-Affected Fruit

The concentration of bacteria and fungi on apple peel after post-harvest storage differed in the two cultivars (Table 3). In GL samples, infections of *V. inaequalis* were associated with a significant increase in microbial populations (one-way ANOVA, *p* < 0.05). In GL_SF samples, bacterial counts were approximately 2 log units, while fungal counts were less than 1 log unit. In contrast, in GL_RT samples, bacterial counts exceeded 6 log units, while fungal counts reached 3 log units per gram of apple. Differences were not significant in GA samples, although a similar tendency was observed: bacterial counts were approximately 2 log units in the SF samples and 3 log units in the RT samples, while fungal counts were higher in RT apples compared to SF fruits (Table 3).

After paired-end alignments, quality filtering, and the deletion of chimeric and singleton sequences, 14,674 reads per sample for fungi, and 15,440 reads per sample for bacteria were obtained and used for downstream analysis. Appendix A present all identified taxa.

The most abundant fungal and bacterial OTUs (contributing more than 0.1% relative abundance) identified in this study were assigned to the phylum and, when possible, to a deeper classification level (Table 4). *Ascomycota* (69.4%) was the dominant fungal phylum, followed by *Basidiomycota* (28.6%); the remaining OTUs were assigned to unclassified fungi (2.0%). Proteobacteria (75.0%) was the dominant bacterial phylum, followed by *Actinobacteria* (10.7%), *Bacteroidetes* (10.8%), and *Firmicutes* (1.1%). The classification of the fungal OTUs resulted in the identification of 140 taxa across the samples (Appendix A).

Bacterial OTUs were assigned to 125 taxa (Appendix A), with *Sphingomonas* being the most abundant (24.8%), followed by *Pseudomonas* (13.1%), *Janthinobacterium* (9.1%), *Methylobacterium* (8.8%), *Hymenobacter* (5.9%), *Microbacteriaceae* family (5.5%), *Acetobacteriaceae* family (5.8%), *Commamonadaceae* family (5.0%), *Enterobacteriaceae* family (3.7%), *Agrobacterium* (2.5%), *Rhodococcus* (2.0%), and *Flavobacterium.* These taxa were distributed at different relative abundances in apple samples based on the presence of apple scab and the specific apple cultivar considered (Figure 4b). *Sphingomonas* was dominant in all samples, with relative abundances between 18.8 and 29.5%. *Acetobacteraceae* (17.4%) and *Enterobacteriaceae* (11.5%) were the most abundant families in the GL_RT samples; however, their abundance decreased dramatically in all other samples, falling to values never exceeding 2.5% and 1.4%. In contrast, some other taxa from the *Proteobacteria* phylum, such as *Commamonadaceae*, *Janthinobacterium*, and *Pseudomonas*, exhibited lower relative abundances in the GL_RT samples while becoming dominant in the other apple peel samples.

Regarding alpha-diversity indices (Table 5), healthy samples of both GL and GA apples exhibited significantly lower richness in bacterial and fungal communities (number of distinct taxa present) compared to scab-infected apples. The Shannon index for bacteria differs only between GA and GL cultivars in the RT fruits, while the same index for fungi significantly differs between SF and RT samples within each cultivar. The evenness index that measures how evenly individuals are distributed across species remained stable in bacteria across both conditions; however, it differed for fungi in SF and RT samples in the GL cultivar.

The beta-diversity distance matrix, which reflects differences in taxa composition between samples based on either presence-absence data or quantitative species abundance, revealed distinct clustering patterns. A cluster analysis of the community structure revealed a higher similarity among the GA samples (RT and SF) and the GL_SF, resulting in two separated clusters: one consisting solely of the GL_RT samples and the other encompassing all the remaining samples for both bacteria and fungi (Figure 5). PERMANOVA analysis confirmed that health status had a significant effect on beta diversity in both fungal (pseudo-*F* = 5.29, *p =* 0.003) and bacterial communities (pseudo-*F* = 3.602, *p =* 0.007).

Analyses at the family and genus levels using ANCOM models revealed several targets for microbiota alterations in both healthy and scab-infected apples, comprising three families and six genera (Table 4), exclusively within the Fungi microbiota.

## 3. Discussion

This study aimed to investigate the potential for valorizing post-harvest scab-infected apples through cider production by comparing the physicochemical properties, fermentation dynamics, cider composition, and peel microbiota of two different apple cultivars. Understanding the interactions between fruit-associated microbiota and fermentation processes is particularly important for optimizing the use of unmarketable apples due to the presence of post-harvest *V. inaequalis* lesions.

The absence of significant differences in physicochemical parameters between healthy and scab-infected apples within each cultivar suggests that *V. inaequalis* infection does not directly compromise the primary quality of the fruits, corroborating previous findings on GL and GA cultivars [27]. Nonetheless, the higher residual sugar and lower ethanol concentration in RT samples from both cultivars indicate a consistent adverse effect of post-harvest scab infection on fermentation efficiency.

Despite the use of a selected *S. cerevisiae* strain to facilitate the alcoholic fermentation of apple juice, scab infections in RT apples negatively affected fermentation performance in the resultant juice, as evidenced by a slower fermentation rate and decreased total sugar consumption (Figure 1). Despite the potentially attainable ethanol levels remaining within the tolerance range for *S. cerevisiae* [18,28], RT apples exhibited impaired fermentative efficiency. This concurs with previous findings indicating that fungal alterations of raw materials can hinder fermentation performance by depleting essential nutrients or generating toxic compounds that directly inhibit the microorganisms responsible for fermentation [29]. The differences in the chemical compositions of the produced ciders corroborate this hypothesis. The partial least squares analysis of volatile and non-volatile compounds in the ciders (Figure 2) highlighted variations in certain substrates that are typically easily assimilated by *S. cerevisiae*, such as hexose sugars and amino acids, as well as in fermentative metabolites, including ethanol, fatty acids, and esters, between SF and RT samples.

An intriguing finding was the shift in sugar metabolism observed in ciders derived from RT apples, where glucose consumption was favored over fructose. This inversion suggests possible yeast stress or altered enzymatic activity during fermentation, as typically seen under suboptimal conditions. The higher residual sugar in the RT samples (Table 1) relative to the SF samples, predominantly composed of glucose, which is a sugar that is typically easily assimilated by *S. cerevisiae* [30], indicates impaired yeast functionality. In addition, the increased fructose assimilation in RT samples suggests competition from other microorganisms, potentially inhibiting the *S. cerevisiae* metabolism [31]. The higher concentration of amino acids in RT ciders further highlights the reduced activity of *S. cerevisiae*, as the inability to assimilate these nitrogen sources not only impairs yeast fermentative activity but also leads to reduced production of fatty acids and esters, which are key aroma compounds derived from yeast metabolism [32]. In addition to the weaker aromatic profile resulting from the poor accumulation of volatile compounds, the composition of ciders produced from RT apples heightens the susceptibility to microbial spoilage due to the presence of residues of readily assimilable substrates (sugars and amino acids).

The PLS and PCA further demonstrate that scab infection significantly affects cider’s chemical composition. RT ciders showed higher levels of amino acids and specific fatty acids, likely derived from increased microbial metabolism or fruit tissue degradation caused by scab infection. Conversely, SF ciders were richer in esters and terpenes, enhancing their sensory properties. This suggests that scab infection could diminish cider aroma complexity, potentially affecting consumer acceptance. The PCA results (Figure 4) indicated that the health status of the apples exerted a greater influence on ciders than cultivar characteristics, with the first principal component (F1) clearly differentiating SF from RT samples. While scab-infected apples remain suitable for cider production, the diversity of their peel microbiota and its impact on fermentation and cider composition cannot be underestimated. To mitigate the negative effect of scab infections and fully exploit the potential of RT apples in cider production, mild antimicrobial technologies such as flash pasteurization of the raw materials, UV or high hydrostatic pressure treatment [33], biocontrol agents [34], and appropriate supplementation of fermentation with growth factors may provide effective solutions.

It is widely recognized that the peel microbiota of fruits plays a crucial role in determining their suitability for human consumption and use as raw materials in the agri-food industry. For example, during post-harvest storage, excessive microbial proliferation on fruits can lead to tissue degradation, uncontrolled fermentations that produce off-flavors, loss of processing quality, and, in severe cases, the accumulation of toxic compounds [35,36,37]. Apple scab typically causes surface lesions without significantly damaging the fruit tissues. However, these peel alterations expose sugars and other nutrients, stimulating microbial growth, as previously observed in other high-quality crops such as wine grapes and berries [25,38]. Our microbial profiling revealed significant shifts in both bacterial and fungal communities associated with apple scab infection and demonstrated that apple scab led to an increase in the microbial population, particularly in GL_RT samples. Varietal characteristics also influenced microbial concentrations on the apple peel, likely due to differences in susceptibility to colonization by *V. inaequalis*, as well as fruit attributes that could stimulate the development of opportunistic microorganisms, including pulp firmness, juiciness, and acidity. The higher microbial load observed in the GL variety (Appendix A) supports this hypothesis.

Metabarcoding analysis of apple peel microbiota confirmed notable differences between scab-infected and healthy apples. The dominant fungal taxa are consistent with those identified in other studies, yet the presence of scab lesions appeared to serve as a reservoir for fungal microbiota. In the fungal community, *Vishniacozyma* and *Alternaria*, known for their protective and antagonistic roles [39], diminished in scab-infected apples, potentially due to competitive exclusion or environmental changes on the fruit surface. Conversely, opportunistic fungi such as *Ramularia* and *Acremonium* increased in the RT environment, suggesting a microbial shift favoring saprophytic or secondary pathogens [40].

Notably, *Vishniacozyma*, *Cladosporium*, *Alternaria*, *Filobasidium Acremonium*, and *Micospherella* were commonly detected, consistent with previous global studies on apple fungal communities [41,42]. Pathogenic fungi, including *Penicillium* [43] and *Fusarium* [44], are causal agents of post-harvest rot. *Penicillium* was detected in all samples, with a higher prevalence in RT samples (GL_RT 1.5% vs. GL_SF 0.1%). *Fusarium* was mainly detected in GL samples (relative abundance consistently exceeding 6%), while in GA apples, it constituted less than 0.5% of microbiota.

*Ramularia mali*, the causal agent of dry lenticel rot [45], was particularly abundant in GL_RT samples, exceeding 30%. Its presence was also detected in SF samples, corroborating previous findings that indicate its population expansion during storage [46]. In GA apples, *Ramularia* was consistently present, albeit at much lower levels, never exceeding 6.5%, regardless of the fruit’s health status, suggesting a greater resistance of this cultivar to the pathogen. While *Ramularia* appears to be an endemic component of the apple epiphytic community, it only seems to cause visible pathogenic effects in susceptible cultivars when its presence exceeds 10%. Our results suggest that GL is more prone to *Ramularia* and *Fusarium* colonization, in accordance with previous observations [47].

White haze, an emerging cosmetic post-harvest apple disorder, is characterized by the imperfection of apple skin [48,49]. It has been associated with yeast-like *basidiomycetes* belonging to the genera *Tilletiopsis*, *Golubevia*, and *Entyloma* (Baric et al., 2010) [50]. In our samples, *Tilletiopsis* was more prevalent on GL samples (2.7 and 1.5% on RT and SF, respectively), consistent with regional observations from northern Italy [51]. The relative abundance of the latter two genera was conversely low and never higher than 0.4%. Although low levels of *Venturia* spp. were detected, the presence of a viable population of *V. inaequalis* can be confirmed by the emergence of post-storage symptoms in both GA and GL samples.

Cultivar and health status significantly impacted both bacterial and fungal diversity, while fungal diversity was more influenced by health status. The marked differences in microbiota composition found in the two cultivars are consistent with previous studies on the cultivar effect [47]. Alpha-diversity analysis revealed richer microbial communities in scab-infected apples, likely due to increased ecological niches from tissue degradation. However, the significant shifts in fungal Shannon diversity between SF and RT apples suggest that scab infection disrupts fungal community evenness, potentially favoring dominance by a few opportunistic taxa. Beta-diversity analysis confirmed distinct clustering patterns based on health status, indicating that scab infection is a primary driver of microbial community shifts, overshadowing the cultivar effect. This aligns with findings from other studies on fruit microbial ecology [25], where disease presence was a dominant factor influencing microbiome composition.

Core bacterial genera such as *Sphingomonas*, *Pseudomonas*, and *Methylobacterium* were consistently detected across samples, aligning with previous findings on apple microbiome [41]. Additionally, *Hymenobacter* was detected, consistent with reports on the Opal cultivar of apples in Italy [46] and the Arlet cultivar in Austria [39]. Certain bacterial taxa, such as *Pseudomonas* and *Janthinobacterium*, include psychotropic species with optimal growth at 10 °C [52] and have previously been found on apple surfaces [53]. Their presence in our samples is probably associated with the prolonged storage of apples at low temperatures. *Sphingomonas*, *Methylobacterium*, and *Pseudomonas* are common inhabitants of the phyllosphere, with several species recognized for their potential as biocontrol agents against foliar and fruit pathogens [54].

Beyond the influence of scab and cultivar on the peel microbiota, it is important to investigate potential correlations between the microbial composition of the apple peel and variations in cider fermentation performance and composition. This aspect is particularly relevant for optimizing the use of RT apples in the cider industry. The ANCOM analysis comparing the microbial communities of SF and RT samples within each cultivar (GL and GA) revealed significant correlations between specific microbial taxa and the presence of scab (Table 4), albeit solely within the fungi community. Several fungi taxa linked to scab infection are known contributors to off-flavor production or fermentation disruption, particularly affecting yeast and lactic acid bacteria activity. It is interesting to note that these correlations were observed across both apple cultivars. The bacterial community exhibited similar trends across SF and RT samples. Some GA-RT samples showed a high relative abundance of *Acetobacteraceae* and *Enterobacteriaceae*, which could influence cider fermentation by producing undesirable metabolites [55]. In contrast, beneficial genera such as *Sphingomonas* and *Pseudomonas*, known for their plant-protective roles [56,57], were present in all the samples. Additionally, the presence of *Cyanobacteria* requires monitoring due to its propensity to produce harmful toxins [58]. Within the fungal community, we observed a substantial absence of Saccharomycetales, the order that includes high-fermentative yeast [59], consistent with observations from other fermentation-related crops, such as grapes [60]. We hypothesized that the spontaneous fermentation of apple cider (without the inoculation of a selected yeast culture) constitutes a risky approach. *Ramularia* and *Vishniacozyma* dominated the fungal community in all the RT samples. Although concrete evidence of their interference with fermentation is lacking, their presence raises concerns. *Ramularia*, already recognized as a pathogen of raw materials intended for fermentation, such as beer malt [61], can compromise product stability and reduce the fermentative potential of vegetables. *Vishniacozyma* is known for its interference with fungal development; therefore, its interactions with *S. cerevisiae* require further clarification to eliminate negative effects on fermentation [62].

## 4. Materials and Methods

### 4.1. Sampling and Storage of Apples

The experiment was repeated on two apple cultivars: Gala (GA) and Golden Delicious (GL). In 2021, GA and GL apples were harvested in two commercial orchards located in the municipality of Predaia (Trentino Alto Adige, Northern, Italy) during the commercial ripening stage between 20–25 August and 10–15 September, respectively. The fruits were stored in a controlled atmosphere at the cold storage facilities of the Consorzio Melinda S.c.A. (Taio, Italy) until 3 March 2022. Storage conditions were maintained at a temperature of 1 °C and relative humidity > 95%. The oxygen levels were 0.8–1% and 1–1.2%, while the carbon dioxide concentrations were 1% and 2.5% for GA and GL, respectively. Integrated pest management was implemented to control pests and diseases in accordance with the guidelines of the “Disciplinare di produzione integrata della Provincia Autonoma di Trento 2021”. For each cultivar, 100 kg of healthy (SF) and scab-infected (RT) apples were randomly selected from the bulk supply. Healthy fruits had no visible scab lesions, while scab-infected fruits displayed 1–5% of their fruit surface covered by post-harvest scab lesions. The samples (GL_SF, GL_RT, GA_SF, and GA_RT) were then transported to the laboratories of the Edmund Mach Foundation in San Michele all’Adige (Italy), for subsequent analysis and experimentation.

### 4.2. Cider Production

For cider production, 30 kg of each apple sample was separately ground with a mechanical blender (MLP0002, Polsinelli Enologia, Isola del Liri, Italy). The resulting pulp was treated with 0.06 g/L of pectolytic enzyme (Endozym Pectofruit PR, AEB Enologia, Brescia, Italy) and 0.05 g/L of sulfur dioxide (Enologica Vason, Verona, Italy). After resting for 12 h at 20 ± 1 °C, the pulp was pressed using a hydraulic press (PRP0046, Polsinelli Enologia), yielding 65% (*w*/*w*) juice. The extracted juice was cold clarified overnight at 6 ± 1 °C, then divided into 5 L glass bottles (three replicates for each sample; Pirex, Vetrotecnica, Vicenza, Italy) and inoculated with *S. cerevisiae* strain ATCC 4228 (American Type Culture Collection, Manassas, VA, USA) at a final concentration of 2.5 × 10^6^ cells/mL. Alcoholic fermentation (AF) was carried out at 22 °C in 10 L glass containers sealed with a non-return valve to enable CO_2_ release. Fermentation progress was monitored daily by measuring weight loss resulting from CO_2_ evolution. Upon completion of AF, the ciders underwent cold stabilization for 7 days at 6 ± 1 °C. Finally, the ciders were decanted and stored in hermetically sealed glass bottles pending chemical analyses.

### 4.3. Physicochemical Analysis of Apples and Ciders

The physicochemical parameters of apples (namely weight, hardness, sugars, acidity, juiciness, and total polyphenols) were measured after cold storage with an automatic fruit-sorting analyzer (Pimprenelle; Setop Giraud Technologie, Cavaillon, France) on 24 apples for each sample. Each cider replicate was chemically characterized at the end of AF. Amino acids were quantified by HPLC (1260 Infinity, Agilent Technologies, Santa Clara, CA, USA) after derivatization with *o*-phthalaldehyde, as reported by Gallo et al. [63]. The system was equipped with a fluorescence detector (Ex = 336 nm, Em = 445 nm) using a Chromolith Performance RP-18e column (100 × 4.6 mm; Merck, Darmstadt, Germany) with a Chromolith RP-18e Guard Cartridge (10 × 4.6 mm; Merck) maintained at 40 °C. Sugars and organic acids were quantified using an ionic chromatograph ICS 5000 (Dionex-Thermo Scientific, Waltham, MA, USA) equipped with an automated eluent generator, a pulsed amperometric detector, and a conductivity cell detector [64]. Mono- and disaccharides were separated on a CarboPac PA200 3 × 250 mm analytical column, preceded by a CarboPac PA200 3 × 50 mm guard column (Dionex-Thermo Scientific) and detected with PAD. Organic acids were analyzed on an IonPac AS11-HC 4 × 250 mm analytical column preceded by an IonPac AS11-HC 4 × 50 mm guard column (Dionex-Thermo Scientific) and detected with COND after a KOH eluent automatic gradient starting at 5 mM that reached 25 mM in 10 min and 66 mM in 1 min. Alcohol was measured by distillation, according to the standard methodology of the International Organization of Vine and Wine [65]. Volatile compounds were quantified in accordance with the method of Paolini et al. [66]. In brief, 50 mL of cider was diluted 1:1 with H_2_O milliQ, and 100 μL of internal standard (n-heptanol, 500 mg/L) was added. Volatiles were extracted by solid-phase extraction using ENV+ cartridges. Gas chromatography-triple quadrupole mass spectrometry (GC-MS/MS) analysis was performed using an Intuvo 9000 system (Agilent Technologies) coupled with a 7000 Series Triple Quadrupole mass spectrometer (Agilent Technologies) operating in electron impact mode at 70 eV. Chromatographic separation was carried out by injecting 2 µL of sample in split mode (1:5) into a DB-Wax Ultra Inert capillary column (20 m, 0.18 mm id × 0.18 µm film thickness) and using helium as a carrier gas (0.8 mL/min). The oven temperature program was as follows: initial hold at 40 °C for 2 min, increase to 55 °C at 10 °C/min, further increase to 165 °C at 20 °C/min and to 240 °C at 40 °C/min, and final hold at this temperature for 5 min. The mass spectrum was acquired in multiple reaction monitoring mode, configuring the instrument in a dynamic system. The injector, transfer line, and ion source temperatures were 260, 250, and 230 °C, respectively.

### 4.4. Quantification of Culturable Microbiota on Apple Peel

The quantification of culturable microbiota on apple peel was carried out using plate counting on 10 fruits (replicates) for each apple sample. Apples were sealed in sterile plastic bags, adding 500 mL of cold (3 °C) sterile water with 0.9% NaCl and 0.1% Tween 20, then shaken for 30 min. The resultant aqueous suspensions (500 mL) were stored at 3 °C until analysis. Three aliquots (1 mL) of each water suspension were subjected to three decimal dilutions, and 0.2 mL was spread over 90-mm Petri dishes containing Potato Dextrose Agar (PDA; Sigma Aldrich, St. Louis, MO, USA) for fungal enumeration and Nutrient Agar (NA; Sigma Aldrich) for bacterial quantification. Petri dishes were incubated at 25 ± 2 °C for a duration of 48 h. The concentrations of fungi and bacteria were referenced to the weight of apples. The calculation and expression of plate count values adhere to UNI EN ISO 7218:2024 [67].

### 4.5. Identification of the Microbiota on Apple Peel

Genomic DNA was extracted from the pellet derived from the apple washing suspensions (three replicates of 25 mL for each apple sample) using the DNeasy PowerFood Microbial Kit (Qiagen, Hilden, Germany). DNA samples were purified using the DNeasy PowerClean Pro Cleanup Kit (Qiagen) and quantified with the Nanodrop 8800 Fluorospectrometer (Thermo Scientific). Amplicon library preparation, quality, and quantification of pooled libraries, and paired-end sequencing utilizing the Illumina MiSeq system (Illumina, San Diego, CA, USA) were conducted at the Sequencing Platform of Fondazione Edmund Mach. For each sample, a 464-nucleotide sequence of the V3–V4 region [68,69] of the 16S rRNA gene (*Escherichia coli* positions 341 to 805) and ITS3/ITS4 specific to the ITS2 fungi region [70] were amplified. Unique barcodes were attached before the forward primers to facilitate the pooling and subsequent differentiation of samples. To prevent preferential sequencing of smaller amplicons, the amplicons were cleaned using the Agencourt AMPure kit (Beckman Coulter, Brea, CA, USA); DNA concentrations of the amplicons were quantified with the Quant-iT PicoGreen dsDNA kit (Invitrogen, Waltham, MA, USA), and the quality of the generated amplicon libraries was evaluated through a Bioanalyzer 2100 (Agilent Technologies) using the High Sensitivity DNA Kit (Agilent Technologies). After quantification, cleaned amplicons were mixed and combined in equimolar ratios. Raw paired-end FASTQ files were demultiplexed using idemp (https://github.com/yhwu/idemp/blob/master/idemp.cpp, accessed on 10 February 2025) and subsequently imported into Quantitative Insights into Microbial Ecology (Qiime2, version 2018.2). Sequences were quality filtered, trimmed, de-noised, and merged using DADA2 [71]. Chimeric sequences were identified and removed via the consensus method in DADA2. Representative bacterial sequences were aligned using MAFFT and employed for phylogenetic reconstruction in FastTree using alignment plugins and phylogeny [72,73]. Taxonomic and compositional analyses for bacteria were conducted using the feature-classifier plugin (https://github.com/qiime2/q2-feature-classifier, accessed on 10 February 2025). A pre-trained Naive Bayes classifier based on the Greengenes 13_8 99% Operational Taxonomic Units (OTUs) database, previously trimmed to the V4 region of 16S rDNA and bound by the 341F/805R primer pair, was applied to paired-end sequence reads to generate taxonomy lists. For fungi, sequences were classified to the species level using a dynamic classifier with thresholds of 97% or 99% created with UNITE software version 8.0 [74]. The data generated by MiSeq Illumina sequencing were deposited in the NCBI Sequence Read Archive and are accessible under Accession No. PRJNA996329.

### 4.6. Statistical Analysis

Significant differences were identified by one-way ANOVA, followed by Tukey’s post-hoc test, after verifying homoscedasticity with Levene’s test and normality through the Shapiro–Wilk test (R Stats Package). Alpha-diversity analysis of Miseq Illumina data was conducted using the number of detected OTUs and the Shannon diversity index; statistical significance between groups was evaluated using the Kruskal–Wallis H test in QIIME2. Beta-diversities were calculated using the Jaccard and Bray–Curtis dissimilarity distance matrices in QIIME2. The output matrix was organized using Principal Coordinate Analysis (PCoA) and visualized using EMPeror [75]. The statistical significance of beta-diversity distances between groups was assessed using PERMANOVA with 999 permutations in QIIME2. The analysis of compositions of microbiomes (ANCOM) was used to uncover differentially abundant features in a microbial dataset, serving as a tool within QIIME2 to compare groups of healthy and scab-infected apples and to identify abundant features, including differential bacteria [76]. ANCOM defines the numbers or ratios represented in the form of a volcano plot.

The correlation between the chemical parameters (non-volatile and volatile constituents) and the apple samples (GL_SF, GL_RT, GA_SF, and GL_RT) was established using Partial Least Squares (PLS) regression and Principal Component Analysis (PCA). PLS and PCA were executed with XLSTAT software (Version 2024.4.2).

## 5. Conclusions

This study contributes to a better understanding of fruit/microbiome interactions and their impact on the quality of fermented apple products, hence reducing food waste and increasing environmental and economic sustainability. The data presented provide valuable insights into the impact of post-harvest apple scab on the peel microbiota of apples and its subsequent effects on cider production, a relatively unexplored aspect. Although scab-infected apples show potential as a raw material for cider production, the observed alterations in the fermentation kinetics and cider compositions cannot be underestimated due to their influence on the overall product quality and chemical profile of ciders. The characterization of the apple peel microbiota revealed that microbial composition is not only influenced by cultivar but, more significantly, by scab infection, despite the disease affecting only a limited area of the fruit surface. This highlights the potential role of scab lesions as microbial reservoirs, which may have important implications for the industrial processing of apples. Given the potential effects of microbial perturbations on fermentation efficiency, it is also necessary to investigate biocontrol options based on microbial interventions to control scab-induced alterations in apple microbiota. Implementing these measures would enhance the sustainability of apple production by reducing food waste and optimizing the industrial processing of scab-infected apples for cider production.

## Figures and Tables

**Figure 1 molecules-30-02322-f001:**
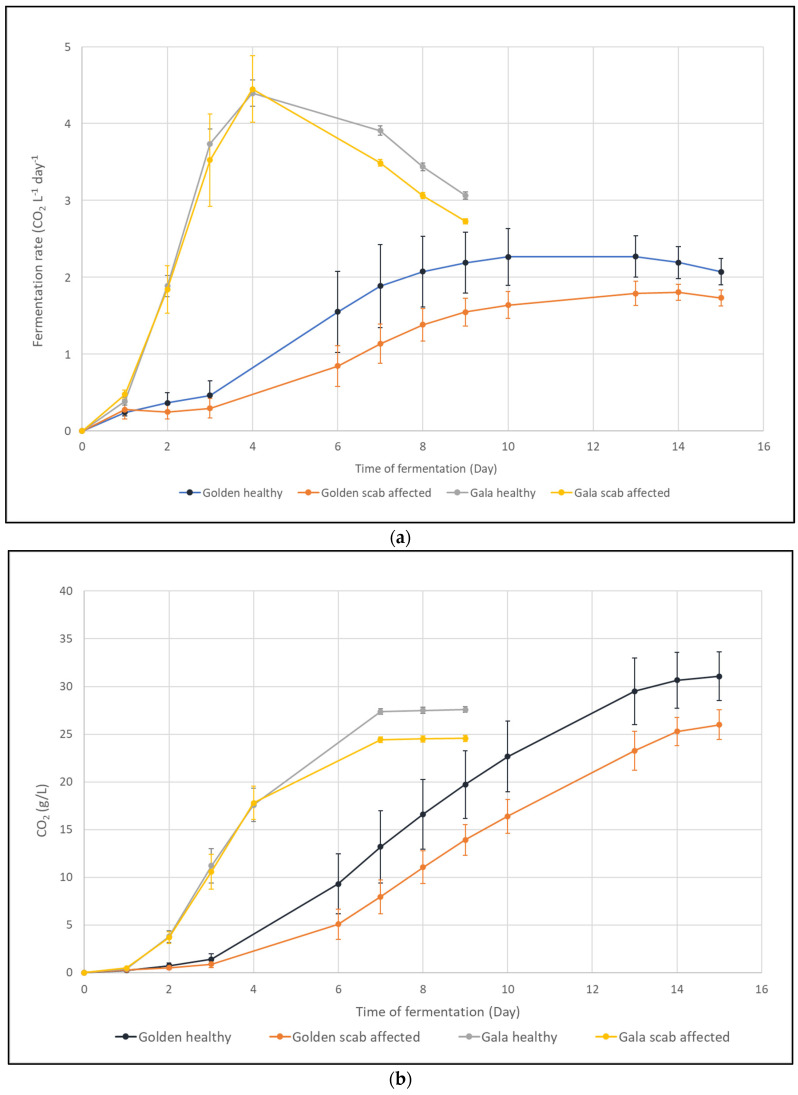
Evolution of alcoholic fermentation in terms of CO_2_ production (**a**) and fermentation rate (**b**) in ciders obtained from samples of healthy or scab-infected Golden Delicious and Gala apples. Mean data ± SD are displayed (n = 3).

**Figure 2 molecules-30-02322-f002:**
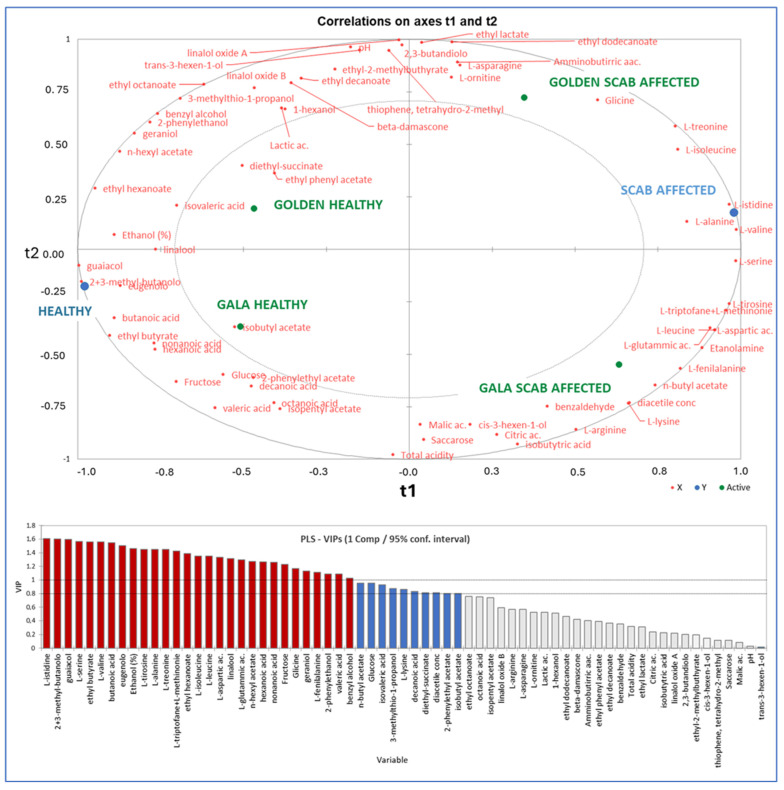
Partial least squares (PLS) regression was carried out on the 60 volatile and non-volatile compounds to identify 31 compounds with a variable importance projection (VIP) score > 1 in ciders produced from both healthy and scab-affected Golden Delicious and Gala apples. Mean data are displayed (n = 3).

**Figure 3 molecules-30-02322-f003:**
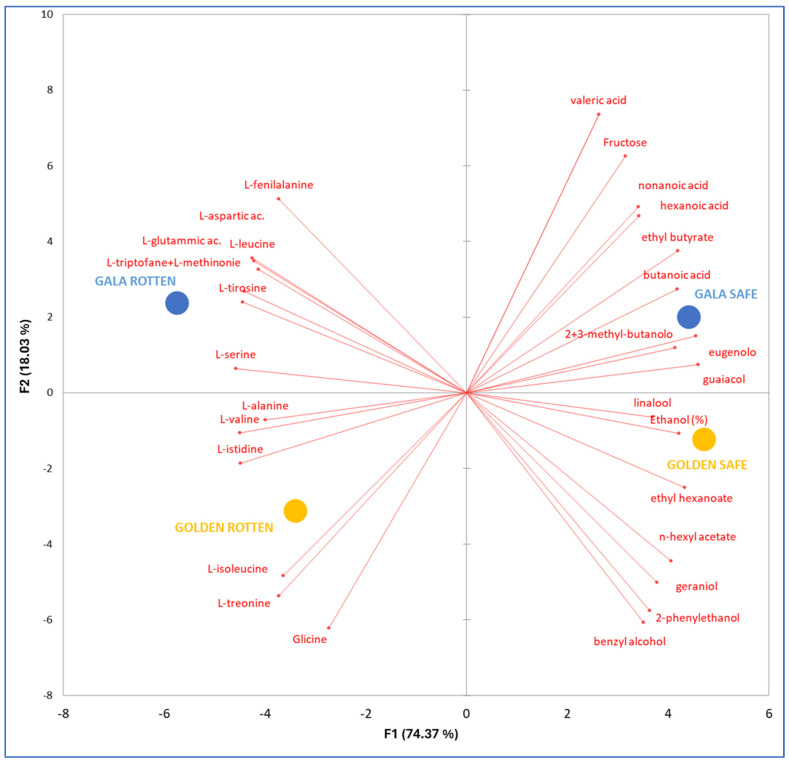
Principal component analysis (PCA) was conducted on the 31 compounds identified in the PLS analysis as having a VIP > 1 in ciders made from both healthy and scab-infected (rotten) Golden Delicious and Gala apples. Projection in the F1/F2 plane of cases and variables. Mean data are displayed (n = 3). Model quality (Comp1): Q^2^ cum: −0.032; R^2^Y cum: 0.962; R^2^X cum: 0.411.

**Figure 4 molecules-30-02322-f004:**
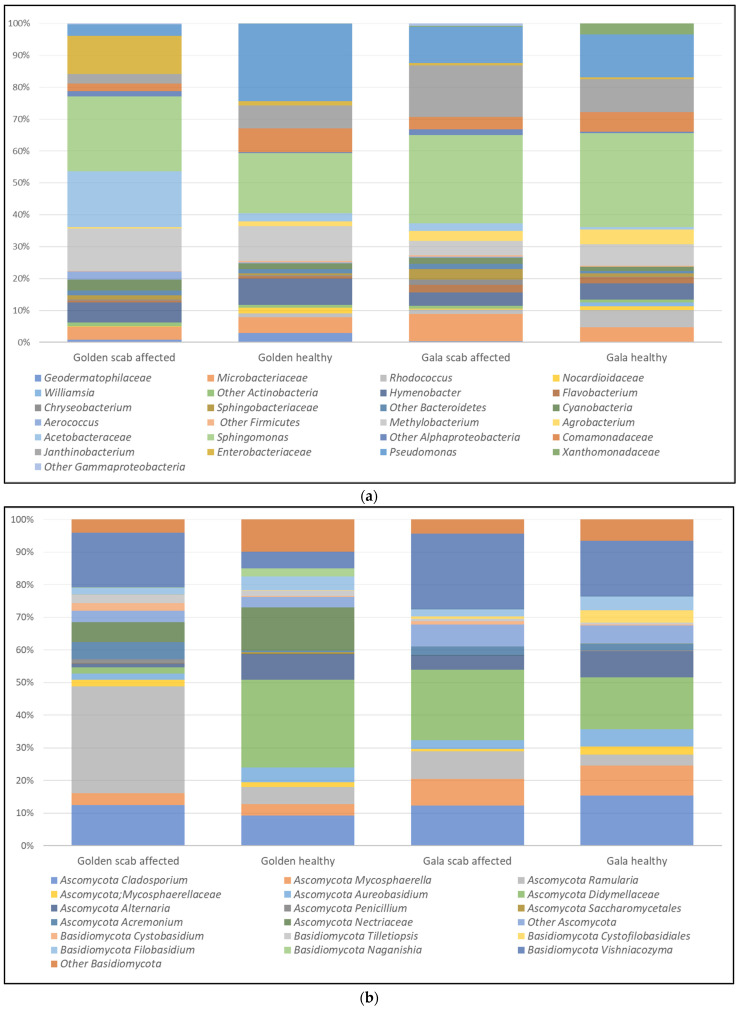
Mean (n = 3) relative abundances of dominant fungal (**a**) and bacterial (**b**) taxa, each contributing more than 0.1%, obtained from the apple peel of healthy and scab-affected Golden Delicious and Gala cultivars after six months of cold storage in a modified atmosphere.

**Figure 5 molecules-30-02322-f005:**
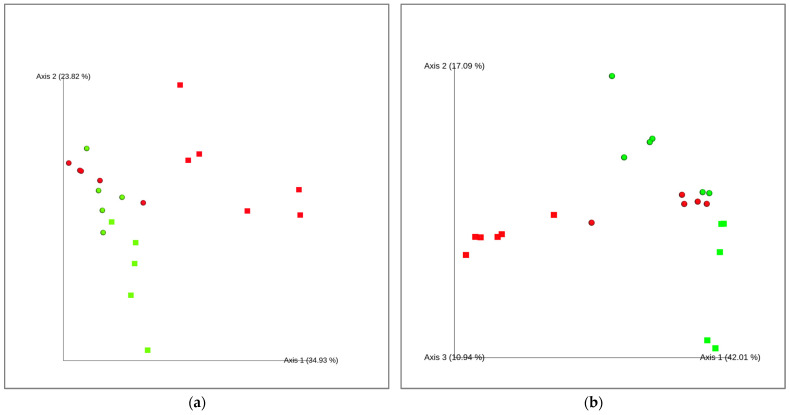
Community structure of bacteria (**a**) and fungi OTUs (**b**) on the peels of Golden Delicious (squares) and Gala apples (circles) in both healthy (green) and scab-affected fruits (red). The complete linkage clustering of the samples was obtained by employing a weighted pair group method with arithmetic mean (UPGMA) based on UNIFRAC distance metrics for bacteria and an unweighted pair group method with arithmetic mean (UPGMA) based on Bray–Curtis distance metrics for fungi.

**Table 1 molecules-30-02322-t001:** Main chemical parameters of ciders (n = 3) obtained from healthy and scab-infected apples from Golden Delicious and Gala cultivars at the end of the alcoholic fermentation. Different letters indicate significant differences within the same apple cultivar (*p* < 0.05, one-way ANOVA and Tukey test).

	Golden Healthy	Golden Scab-Affected	Gala Healthy	Gala Scab-Affected
pH	3.88 ± 0.10 ^a^	3.94 ± 0.13 ^a^	3.84 ± 0.01 ^a^	3.78 ± 0.02 ^b^
Citric acid (g/L)	0.1 ± 0.5 ^a^	0.1 ± 0.3 ^a^	0.1 ± 0.1 ^a^	0.1 ± 0.0 ^a^
Malic acid (g/L)	1.7 ± 2.1 ^a^	1.2 ± 1.9 ^a^	5.3 ± 0.6 ^a^	4.4 ± 0.1 ^b^
Lactic acid (g/L)	0.9 ± 1.4 ^a^	0.9 ± 0.7 ^a^	0.1 ± 0.1 ^a^	0.1 ± 0.3 ^a^
Fructose (mg/L)	36.5 ± 1.2 ^a^	29.2 ± 3.6 ^b^	52.7 ± 7.8 ^a^	36.2 ± 7.7 ^b^
Glucose (mg/L)	19.4 ± 3.5 ^a^	36.3 ± 1.2 ^b^	9.3 ± 5.6 ^a^	46.7 ± 7.7 ^b^
Sucrose (mg/L)	3.2 ± 2.1 ^a^	3.1 ± 2.9 ^a^	12.6 ± 6.4 ^a^	11.5 ± 5.6 ^a^
Ethanol (%)	7.5 ± 0.1 ^a^	7.1 ± 1.4 ^b^	7.3 ± 0.1 ^a^	7.1 ± 0.0 ^b^

**Table 2 molecules-30-02322-t002:** Quantification of volatile and non-volatile compounds of ciders obtained from healthy and scab-infected apples of Golden Delicious and Gala cultivars at the end of the alcoholic fermentation. Mean data (n = 3) and the variable importance (VIP) score derived from the partial least squares (PLS) model.

Sample	Gala Healthy	SD	Gala Scab-Affected	SD	Golden Healthy	SD	Golden Scab-Affected	SD	VIP
			Volatile compounds (mg/L)						
Isobutyl Acetate	0.007	0.001	0.004	0.001	0.004	0.001	0.004	0.001	0.802
Ethyl Butyrate	0.044	0.009	0.034	0.004	0.042	0.003	0.030	0.008	1.561
Ethyl-2-methylbutyrate	0.002	0.001	0.002	0.000	0.018	0.002	0.015	0.003	0.191
n-Butyl Acetate	1.326	0.124	1.658	0.021	1.293	0.075	1.318	0.038	0.957
Isopentyl Acetate	0.265	0.087	0.153	0.014	0.095	0.009	0.067	0.007	0.739
Ethyl Hexanoate	0.190	0.006	0.092	0.008	0.186	0.027	0.148	0.016	1.388
n-Hexyl Acetate	0.014	0.001	0.010	0.001	0.016	0.001	0.013	0.002	1.270
Ethyl Lactate	0.169	0.035	0.141	0.017	8.681	2.277	12.974	1.310	0.310
1-Hexanol	1.407	0.040	1.339	0.050	4.515	0.400	2.923	1.054	0.513
trans-3-Hexen-1-ol	0.022	0.002	0.017	0.001	0.060	0.015	0.064	0.005	0.015
cis-3-Hexen-1-ol	0.026	0.002	0.025	0.003	0.004	0.001	0.009	0.004	0.146
Ethyl Octanoate	0.058	0.011	0.021	0.005	0.076	0.012	0.072	0.012	0.756
Linalool Oxide A	0.003	0.002	0.002	0.000	0.008	0.001	0.012	0.002	0.222
Linalool Oxide B	0.003	0.002	0.001	0.000	0.008	0.001	0.006	0.000	0.595
Benzaldehyde	0.016	0.000	0.029	0.002	0.019	0.001	0.012	0.002	0.356
Isobutyric acid	0.079	0.014	0.091	0.002	0.073	0.003	0.066	0.012	0.225
Linalool	0.024	0.001	0.021	0.001	0.032	0.003	0.019	0.002	1.317
Butanoic acid	0.351	0.003	0.292	0.078	0.368	0.019	0.263	0.091	1.549
Ethyl Decanoate	0.012	0.000	0.007	0.003	0.054	0.009	0.041	0.013	0.368
Isovaleric Acid	0.342	0.015	0.206	0.006	0.264	0.055	0.288	0.025	0.927
Diethyl-Succinate	0.004	0.000	0.003	0.000	0.050	0.004	0.012	0.004	0.815
3-Methylthio-1-Propanol	1.202	0.024	0.149	0.085	1.510	0.150	1.392	0.030	0.875
Valeric Acid	0.025	0.001	0.021	0.004	0.021	0.001	0.018	0.002	1.086
Ethyl Phenyl Acetate	0.009	0.000	0.006	0.001	0.006	0.002	0.008	0.001	0.393
2-Phenylethyl Acetate	0.038	0.008	0.016	0.002	0.010	0.004	0.009	0.001	0.802
β-Damascone	0.019	0.001	0.017	0.001	0.028	0.003	0.025	0.008	0.423
Hexanoic Acid	1.139	0.036	0.688	0.035	0.799	0.091	0.629	0.185	1.267
Ethyl Dodecanoate	0.102	0.023	0.104	0.007	0.144	0.023	0.177	0.008	0.467
Geraniol	0.318	0.036	0.188	0.068	0.337	0.021	0.303	0.014	1.134
Guaiacol	0.007	0.001	0.004	0.002	0.007	0.000	0.005	0.001	1.598
Benzyl Alcohol	0.218	0.008	0.098	0.009	0.275	0.031	0.228	0.046	1.028
2-Phenylethanol	20.270	0.677	15.888	0.695	22.739	1.701	20.428	1.032	1.088
Octanoic Acid	1.707	0.214	1.150	0.123	0.893	0.155	0.780	0.170	0.754
Nonanoic Acid	0.008	0.002	0.003	0.001	0.005	0.001	0.003	0.000	1.260
Decanoic Acid	0.433	0.119	0.317	0.057	0.285	0.080	0.269	0.017	0.832
Diacetyl	6.227	2.014	7.653	0.196	5.975	0.257	5.964	0.175	0.815
3-Methyl-2-butanol	60.365	1.573	52.434	2.848	59.376	2.449	53.082	4.611	1.600
2,3-Butanediol	0.750	0.005	0.636	0.023	5.786	6.277	7.399	0.744	0.201
Thiophene. Tetrahydro-2-Methyl	1.532	0.390	1.544	0.033	2.339	0.126	2.451	0.035	0.115
Eugenol	0.544	0.045	0.222	0.006	0.794	0.093	0.102	0.004	1.502
		Non-volatile compounds (g/L)							
Aminobutyric acid	0.333	0.122	0.433	0.117	0.633	0.221	0.675	0.121	0.405
L-aspartic acid	0.533	0.045	1.500	0.068	0.450	0.028	0.850	0.098	1.333
L-glutamic acid	0.500	0.023	1.933	0.021	0.575	0.036	0.975	0.078	1.294
L-alanine	1.067	0.087	1.300	0.045	0.800	0.021	1.400	0.121	1.450
L-arginine	0.867	0.122	1.500	0.138	0.300	0.184	0.350	0.089	0.568
L-asparagine	0.367	0.125	0.300	0.107	0.350	0.215	0.550	0.068	0.567
Ethanolamine	3.733	0.064	5.600	0.184	3.375	0.129	4.150	0.098	1.360
L-phenylalanine	0.333	0.022	0.767	0.092	0.275	0.027	0.375	0.111	1.111
Glycine	0.900	0.048	1.233	0.141	0.700	0.148	2.975	0.210	1.167
L-isoleucine	0.167	0.066	0.367	0.021	0.300	0.018	0.400	0.084	1.352
L-histidine	0.233	0.045	0.433	0.058	0.225	0.031	0.450	0.066	1.608
L-leucine	0.600	0.022	1.133	0.031	0.500	0.086	0.775	0.054	1.350
L-lysine	1.133	0.102	1.967	0.122	0.475	0.086	0.825	0.098	0.863
L-ornithine	0.600	0.148	0.467	0.089	0.525	0.128	0.850	0.036	0.528
L-serine	1.067	0.022	2.267	0.076	0.825	0.084	1.950	0.122	1.565
L-tyrosine	0.333	0.096	0.633	0.048	0.300	0.052	0.475	0.046	1.451
L-threonine	0.300	0.086	0.467	0.081	0.350	0.021	0.600	0.084	1.450
L-tryptophan + L-methionine	0.333	0.054	0.600	0.047	0.300	0.032	0.450	0.096	1.428
L-valine	0.233	0.048	0.533	0.032	0.300	0.048	0.475	0.021	1.557

**Table 3 molecules-30-02322-t003:** Concentration of culturable bacteria and fungi on the peels of healthy and scab-affected apples from the two cultivars (Gala and Golden Delicious). Mean ± SD (n = 3). Statistically significant differences in the same apple cultivar are indicated by different letters in superscript (*p* < 0.05, one-way ANOVA).

Apple Sample	Bacteria	Fungi
	×10^3^ CFU/g	
Golden healthy	2.3 ± 1.2 ^a^	0.6 ± 0.4 ^a^
Golden scab-affected	6.1 ± 1.7 ^b^	2.8 ± 0.8 ^b^
Gala healthy	9.1 ± 1.4 ^a^	3.2 ± 1.2 ^a^
Gala scab-affected	10 ± 0.7 ^a^	4.5 ± 1.6 ^a^

**Table 4 molecules-30-02322-t004:** Taxa composition (expressed as a percentage) in both healthy and scab-affected Gala and Golden Delicious apple fruit samples as revealed by high-throughput sequencing analysis. The gray squares represent relative abundances in the range of 1–10%, while the dark gray squares represent relative abundances in the range of 11–100%. The star denotes taxa that exhibit a significant difference between the SF and RT apple samples within each cultivar, as determined by the analysis of compositions of microbiomes (ANCOM). n.d.: Not detectable.

Bacteria (Relative Abundance)				
	Gala Healthy	Gala Scab-Affected	Golden Healthy	Golden Scab-Affected
*Geodermatophilaceae*	0.395	n.d.	0.740	2.825
*Microbacteriaceae*	8.510	4.687	4.057	4.917
*Rhodococcus*	1.208	5.430	n.d.	1.262
*Nocardioidaceae*	0.334	1.162	0.183	1.840
*Williamsia*	n.d.	1.130	n.d.	n.d.
*Other Actinobacteria*	0.894	0.964	1.223	0.889
*Hymenobacter*	4.200	5.113	6.172	8.113
*Flavobacterium*	2.340	1.920	0.603	0.740
*Chryseobacterium*	1.722	0.213	0.393	0.390
*Sphingobacteriaceae*	3.152	0.920	1.298	0.553
*Other Bacteroidetes*	1.703	0.849	1.499	1.329
*Cyanobacteria*	1.911	1.304	3.434	1.803
*Aerococcus*	0.251	n.d.	2.519	0.258
*Other Firmicutes*	0.472	0.334	0.114	0.403
*Methylobacterium*	4.455	6.775	13.264	10.875
*Agrobacterium*	3.235	4.526	0.484	1.608
*Acetobacteraceae*	2.357	0.813	17.415	2.484
*Sphingomonas*	27.560	29.487	23.394	18.762
*Other Alphaproteobacteria*	1.665	0.406	1.628	0.354
*Comamonadaceae*	3.949	6.192	2.263	7.431
*Janthinobacterium*	16.103	10.259	3.083	7.073
*Enterobacteriaceae*	0.678	0.623	11.942	1.404
*Pseudomonas*	11.321	13.465	3.476	24.095
*Xanthomonadaceae*	0.260	3.407	n.d.	0.018
*Other Gammaproteobacteria*	0.756	0.020	0.307	0.142
**Fungi (relative abundance)**				
	**Gala healthy**	**Gala scab-affected**	**Golden healthy**	**Golden scab-affected**
*Ascomycota*; *Cladosporium*	12.430	9.285	12.263	15.358
*Ascomycota*; *Mycosphaerella*	3.701	3.397	8.204	9.243
*Ascomycota*; *Ramularia **	32.646	2.355	6.503	3.376
*Ascomycota*; *Mycosphaerellaceae*	2.028	1.449	0.679	2.445
*Ascomycota*; *Aureobasidium*	1.975	4.438	2.647	5.285
*Ascomycota*; *Didymellaceae **	1.831	26.948	21.655	15.838
*Ascomycota*; *Alternaria **	1.004	7.848	4.295	8.084
*Ascomycota*; *Penicillium*	1.489	0.038	0.031	0.217
*Ascomycota*; *Saccharomycetales*;	n.d.	0.500	0.022	n.d.
*Ascomycota*; *Acremonium*	4.204	0.461	2.586	1.888
*Ascomycota*; *Nectriaceae **	6.155	13.263	0.087	0.228
*Other Ascomycota*	3.575	3.247	6.781	5.525
*Basidiomycota*; *Cystobasidium **	2.275	0.330	1.024	0.293
*Basidiomycota*; *Tilletiopsis*	2.647	1.532	0.722	0.661
*Basidiomycota*; *Cystofilobasidiales*	0.051	0.206	0.713	3.785
*Basidiomycota*; *Filobasidium **	2.056	4.290	2.303	4.201
*Basidiomycota*; *Naganishia **	0.135	1.784	0.011	0.015
*Basidiomycota*; *Vishniacozyma **	15.765	5.090	24.183	17.060
*Other Basidiomycota*	4.189	8.889	4.811	5.549

**Table 5 molecules-30-02322-t005:** Comparison of alpha-diversity indices (operational taxonomic units [OTUs], Shannon, and Evenness) across microbial communities on the peels of scab-affected and healthy Golden Delicious and Gala apple cultivars. The different letters denote significant differences (*p* < 0.05) in mean values (n = 3) as determined by the Kruskal–Wallis test.

Sample	Observed OTUs	Shannon Index	Evenness Index
Bacteria			
Golden healthy	69 ± 6 ^a^	5.496 ± 0.378 ^ab^	0.890 ± 0.029 ^a^
Golden scab-affected	91 ± 25 ^b^	5.690 ± 0.520 ^b^	0.885 ± 0.026 ^a^
Gala healthy	58 ± 17 ^a^	5.411 ± 0.495 ^ab^	0.898 ± 0.027 ^a^
Gala scab-affected	155 ± 68 ^b^	5.362 ± 0.472 ^a^	0.893 ± 0.022 ^a^
Fungi			
Golden healthy	74 ± 5 ^b^	4.502 ± 0.283 ^b^	0.707 ± 0.042 ^a^
Golden scab-affected	87 ± 14 ^c^	4.617 ± 0.232 ^c^	0.719 ± 0.025 ^b^
Gala healthy	61 ± 10 ^a^	4.391 ± 0.257 ^a^	0.711 ± 0.049 ^ab^
Gala scab-affected	74 ± 11 ^b^	4.462 ± 0.290 ^b^	0.708 ± 0.040 ^a^

## Data Availability

Data are contained within the article and Appendix A.

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
