# Peer review of "Impact of Post-Harvest Apple Scab on Peel Microbiota, Fermentation Dynamics, and the Volatile/Non-Volatile Composition of Cider"

_molecules, 2025, doi:10.3390/molecules30112322_

Round 1
Reviewer 1 Report
Comments and Suggestions for Authors
The authors conducted the research to generate an understanding of the alcoholic beverage cider and evaluated what happens to the natural microbiota of apples if they are contaminated by the fungus Venturia inaequalis? Since there is economic loss in the crop and in postharvest.
The analytical methodology to evaluate the physicochemical parameters of Brix, ph, malic acid content, citric, alcohol content is correct according to the AOAC that I had the opportunity to review, as well as the results obtained in the research and that coincide with what is reported in the cited literature.
The special HPLC analysis is fundamental and correct, they found 60 volatile and non-volatile compounds according to other investigations.
They also elucidate what happens with the specific of ochratoxin, its cycle, and specify that although it does not intervene in the microbiota to carry out alcoholic fermentation, they interpret that it can be processed as part of the circular economy to take advantage of the apple contaminated by Venturia inaequalis, because it does not affect the microbiota.
From the ethical point of view that I did not emit is that they should not be used because of the damage they can cause to human health, although cider is not of frequent or daily consumption as some other beverage of high consumption.
The statistical analysis is adequate.
That is why I did not issue further judgment and observation to the paper. In my experience the paper may be publishable as science popularization.
Author Response
Thanks to Reviewer 1 for the positive comments on our work. Regarding the form of the English language, we have requested a complete revision of the text by a professional reviewer, I hope it will be more readable and correct.
Best regards.
Raffaele Guzzon
Reviewer 2 Report
Comments and Suggestions for Authors
In this work apples affected by post-harvest apple scab were employed in cidermaking, in comparison with safe apples. The peel microbiota of apples was described by plate count and next generation sequencing based metabarcoding approaches and the composition of ciders was characterized by HPLC and GC MS-MS to establish the suitability of affected fruits in cider industry. The microbiota results linkable to the sanitary state of fruits, apple affected by post-harvest scab host a rich and peculiar fungal consortium.In generally, the result were well shown. I suggest accept it after revisions.
- Abstract: I suggest add some data to abstract. Present abstract was too generally.
- Table 1, the contents of Glucose were increased after scab affected, please discuss it in text.
- Table 2, how many repeats? I suggest the standard deviation should be added to data.
- Figure 2.P value?
- Table 3. 2.8 ± 3.8, This value is not scientific, I suggest the author redo it.
- Table 4 and Figure 4, how many repeats?
- Figure 5.What do different colors and shapes represent? There should be markings in the picture.
- Part 5. Conclusions, I Suggest delete the 2nd paragraph.
- Figure captions. In each Figure, there are too many abbreviations. I suggest the Abbreviation should be full names in figure captions.
- The writing English should be improved by native English speaker. I strongly suggest to check the manuscript carefully, especially please check the grammar and the completeness of the sentences once again. And please check the tense of the sentences. There should be consistency throughout the manuscript using past tense.
Comments on the Quality of English Language
In this work apples affected by post-harvest apple scab were employed in cidermaking, in comparison with safe apples. The peel microbiota of apples was described by plate count and next generation sequencing based metabarcoding approaches and the composition of ciders was characterized by HPLC and GC MS-MS to establish the suitability of affected fruits in cider industry. The microbiota results linkable to the sanitary state of fruits, apple affected by post-harvest scab host a rich and peculiar fungal consortium.In generally, the result were well shown. I suggest accept it after revisions.
- Abstract: I suggest add some data to abstract. Present abstract was too generally.
- Table 1, the contents of Glucose were increased after scab affected, please discuss it in text.
- Table 2, how many repeats? I suggest the standard deviation should be added to data.
- Figure 2.P value?
- Table 3. 2.8 ± 3.8, This value is not scientific, I suggest the author redo it.
- Table 4 and Figure 4, how many repeats?
- Figure 5.What do different colors and shapes represent? There should be markings in the picture.
- Part 5. Conclusions, I Suggest delete the 2nd paragraph.
- Figure captions. In each Figure, there are too many abbreviations. I suggest the Abbreviation should be full names in figure captions.
- The writing English should be improved by native English speaker. I strongly suggest to check the manuscript carefully, especially please check the grammar and the completeness of the sentences once again. And please check the tense of the sentences. There should be consistency throughout the manuscript using past tense.
Author Response
Thanks for the valuable suggestions, the changes in the text are highlighted in yellow. In detail here are the answers to the comments of Referee 2.
Abstract: I suggest add some data to abstract. Present abstract was too generally.
Response. Abstract was improved please see the part underlined in yellow in the text.
Table 1, the contents of Glucose were increased after scab affected, please discuss it in text.
Response. This effect is probably due at the different composition of population and at the minor activity of S. cerevisiae that, as is largely know, prefers glucose in alcoholic fermentation. Some considerations were added in the text from line 288.
Table 2, how many repeats? I suggest the standard deviation should be added to data.
Response. Data are mean of 3 replicates, please see the caption of Table 2 (Line 126). The authors did not add the Ds because it would greatly complicate the readability of the table and because the statistical approach that was applied considers the entire profile of the compounds analysed. It is evident that in cases where there is a statistical difference in a compound between the theses, this exceeds the standard deviation within the 3 replicates of each thesis.
Figure 2.P value?
REsponse. Statistical parmaters were added in the figure caption.
Table 3. 2.8 ± 3.8, This value is not scientific, I suggest the author redo it.
Response. The data in Table 3 are expressed with two significant figures, as required by international standards in the sector (e.g. ISO 7218), indicating Mean (n=3) and Standard deviation (SD). I believe this is the correct way to express microbial concentrations in food samples. If the reviewer has any other suggestions, they are appreciated.
Table 4 and Figure 4, how many repeats?
Response. Table 4 and figure 4 refer to the same dataset but provide different information. Table 4 shows all identified taxa and the results of the ANCOM statistical analysis. Figure 4 clearly displays the most significant taxa. However, if the editor belives it appropriate, we can eliminate Figure 4. I await a response about this point.
Figure 5. What do different colours and shapes represent? There should be markings in the picture.
Response. Colour and shapes indicate the different samples, please see figure caption that was improved.
Part 5. Conclusions, I Suggest delete the 2nd paragraph.
Response. The 2nd paragraph was delated and the conclusion fully reconsidered.
Figure captions. In each Figure, there are too many abbreviations. I suggest the Abbreviation should be full names in figure captions.
Response. Figure captions were improved.
The writing English should be improved by native English speaker. I strongly suggest to check the manuscript carefully, especially please check the grammar and the completeness of the sentences once again. And please check the tense of the sentences. There should be consistency throughout the manuscript using past tense.
Answer. English was improved by a professional service of translation.
Reviewer 3 Report
Comments and Suggestions for Authors
Impact of post-harvest apple scab on peel microbiota, fermentation dynamics and cider volatile/non-volatile composition
This is an interesting article. However, I was really able to capture the main goal after this statement in the introduction “we carried out a thorough chemical characterization of cider produced from both healthy and scab-affected fruits, to identify compositional changes caused by microbial proliferation and to assess their impact on cider quality.” I think this statement should be stated early on in the abstract.
The font sizes in the figures are too small and hard to read.
Table 1 looks very crowded.
Figure 4 is also hard to read as well.
Table 5. Please provide meaning of OUT acronym.
I think overall this work is really important since it explores the impact of post-harvest apple scab on the peel microbiota of apples and its subsequent effects on cider production. I think this manuscript just requires clarification of the goals and novelty.
Author Response
Thanks for the valuable suggestions, the changes in the text are highlighted in yellow. In detail here are the answers to the comments of Referee 3.
This is an interesting article. However, I was really able to capture the main goal after this statement in the introduction “we carried out a thorough chemical characterization of cider produced from both healthy and scab-affected fruits, to identify compositional changes caused by microbial proliferation and to assess their impact on cider quality.” I think this statement should be stated early on in the abstract.
Response. The abstract was improved as suggested by Reviewer 3, please see the part underlined in yellow.
The font sizes in the figures are too small and hard to read.
Response. Figure were improved where is possible because in some cases are generated by statistical programs without possibility of modification.
Table 1 looks very crowded.
Response. Table 1 was fully redesigned to allow better readability.
Figure 4 is also hard to read as well.
Response Figure 4 was improved.
Table 5. Please provide meaning of OUT acronym.
Response. The information was inserted in figure caption.
I think overall this work is really important since it explores the impact of post-harvest apple scab on the peel microbiota of apples and its subsequent effects on cider production. I think this manuscript just requires clarification of the goals and novelty.
Response. The authors have undertaken a thorough revision of the English language to clarify the purposes of the manuscript.
Reviewer 4 Report
Comments and Suggestions for Authors
This study evaluates the use of post-harvest apple scab-affected apples in cidermaking. However:
- While the Authors state in the discussion section that the aim of the study was to explore the potential for valorizing post-harvest scab-affected apples through cider production, this aim is neither clearly presented in the abstract nor in the introduction. Moreover, although the conclusion section summarizes interesting findings related to peel microbiota, fermentation kinetics, and cider composition, it does not explicitly link the results back to the stated aim of the study. As a result, there is a lack of clear closure and coherence between the aim, results, and conclusions. I recommend that the Authors revise the abstract and introduction to clearly present the research aim, and that the conclusion be restructured to directly address whether and how the aim of the study was achieved.
- A significant proportion of the references cited in the manuscript are older than five years, which is unusual for a research article aiming to address an evolving and technologically sensitive topic. The Authors should integrate and discuss more recent findings from the literature to ensure the study is positioned within the current state of knowledge.
- Tables 2 and TS1 present data without any accompanying statistical information, such as measures of variability (e.g., standard deviation, standard error) or significance testing results.
- To enhance the transparency and completeness of the presented data, it would be valuable if the Authors could provide the HPLC chromatograms of their analyses as part of the supplementary material.
Author Response
Thanks for the valuable suggestions, the changes in the text are highlighted in yellow. In detail here are the answers to the comments of Referee 4.
While the Authors state in the discussion section that the aim of the study was to explore the potential for valorizing post-harvest scab-affected apples through cider production, this aim is neither clearly presented in the abstract nor in the introduction. Moreover, although the conclusion section summarizes interesting findings related to peel microbiota, fermentation kinetics, and cider composition, it does not explicitly link the results back to the stated aim of the study. As a result, there is a lack of clear closure and coherence between the aim, results, and conclusions. I recommend that the Authors revise the abstract and introduction to clearly present the research aim, and that the conclusion be restructured to directly address whether and how the aim of the study was achieved.
Response. According to the suggestion of different reviewers, the abstract and the introduction were revised, considering the limit in number of words imposed by the standard of the journal.
A significant proportion of the references cited in the manuscript are older than five years, which is unusual for a research article aiming to address an evolving and technologically sensitive topic. The Authors should integrate and discuss more recent findings from the literature to ensure the study is positioned within the current state of knowledge.
Response. Old references are useful, in the opinion of authors, because they are milestone in the field treated of this work. Also, for some specific themes it is not always possible to find recent articles. In any case we have tried to improve the bibliographic section by replacing, where possible, the references with more modern references.
Tables 2 and TS1 present data without any accompanying statistical information, such as measures of variability (e.g., standard deviation, standard error) or significance testing results.
To enhance the transparency and completeness of the presented data, it would be valuable if the Authors could provide the HPLC chromatograms of their analyses as part of the supplementary material, please see Figure S1.
Round 2
Reviewer 2 Report
Comments and Suggestions for Authors
Table 3. 2.8 ± 3.8, authors explained Mean (n=3) and Standard deviation (SD). However, it is impossible Mean more than Standard deviation (SD).
Author Response
Comment: Table 3. 2.8 ± 3.8, authors explained Mean (n=3) and Standard deviation (SD). However, it is impossible Mean more than Standard deviation (SD).
Response: Thanks for the suggestion. This is a transcription error in the table. The data has been updated and the statistical analysis repeated to verify that the difference is significant. Find the correction highlighted in purple in revision 2 of the manuscript.
Reviewer 4 Report
Comments and Suggestions for Authors
Comments 1 and 2 have been sufficiently addressed. The revised abstract and introduction now better reflect the stated aim of the study, and the updates to the references demonstrate an effort to incorporate more recent literature where appropriate.
In contrast, comments 3 and 4 have not been addressed at all. No response or justification was provided regarding the absence of statistical information in Tables 2 and TS1, nor was there any mention of the requested HPLC chromatograms in the supplementary materials.
Author Response
Dear editor, dear Revisor 4.
We have carefully read the comments of reviewer 4. Regarding the lack of inclusion of the HPLC chromatograms, please note that these were already attached to revision 1 of the manuscript, as figures in the supplementary materials. Also, you can find the link in the text at line 121 of page 4 and line 572 of page 18 (Text underlined in purple).
In Table 2 we have added the DS (Text and data underlined in purple).
Finally, for the data in Table S1, we used the Analysis of Microbiome Composition (ANCOM) to assess differences between different microbial groups (see lines 140-143). ANCOM is a differential abundance (DA) analysis for microbial absolute abundances. It takes into account the compositionality of the microbiome data by performing the additive log ratio (LRR) transformation. ANCOM looks for differences in taxa between different cases: in our study of healthy and scab-infected apples we cannot use ANCOM to see differences between taxa without considering cases. In our study of healthy and scab-infected apples, ANCOM results are shown in Table 4, where Table S1 shows the full set of identified taxa, as a response to another reviewer's request.